# Unlocking Therapeutic Synergy: Tailoring Drugs for Comorbidities such as Depression and Diabetes through Identical Molecular Targets in Different Cell Types

**DOI:** 10.3390/cells12232768

**Published:** 2023-12-04

**Authors:** Thierry Coppola, Guillaume Daziano, Ilona Legroux, Sophie Béraud-Dufour, Nicolas Blondeau, Patricia Lebrun

**Affiliations:** CNRS, IPMC, Université Côte d’Azur, Sophia Antipolis, F-06560 Valbonne, France; guillaume.daziano@gmail.com (G.D.); legroux@ipmc.cnrs.fr (I.L.); beraud@ipmc.cnrs.fr (S.B.-D.); blondeau@ipmc.cnrs.fr (N.B.)

**Keywords:** depression, diabetes, pharmacology, cell signaling, receptor, channel

## Abstract

Research in the field of pharmacology aims to generate new treatments for pathologies. Nowadays, there are an increased number of chronic disorders that severely and durably handicap many patients. Among the most widespread pathologies, obesity, which is often associated with diabetes, is constantly increasing in incidence, and in parallel, neurodegenerative and mood disorders are increasingly affecting many people. For years, these pathologies have been so frequently observed in the population in a concomitant way that they are considered as comorbidities. In fact, common mechanisms are certainly at work in the etiology of these pathologies. The main purpose of this review is to show the value of anticipating the effect of baseline treatment of a condition on its comorbidity in order to obtain concomitant positive actions. One of the implications would be that by understanding and targeting shared molecular mechanisms underlying these conditions, it may be possible to tailor drugs that address both simultaneously. To this end, we firstly remind readers of the close link existing between depression and diabetes and secondly address the potential benefit of the pleiotropic actions of two major active molecules used to treat central and peripheral disorders, first a serotonin reuptake inhibitor (Prozac ^®^) and then GLP-1R agonists. In the second part, by discussing the therapeutic potential of new experimental antidepressant molecules, we will support the concept that a better understanding of the intracellular signaling pathways targeted by pharmacological agents could lead to future synergistic treatments targeting solely positive effects for comorbidities.

## 1. Introduction

It is generally accepted that a drug’s pleiotropy is observed when its effects are different from those initially intended. This includes both negative and positive effects. Up to now, research into and the development of treatments for a given pathology have generated drugs which, when used, have revealed multiple side effects, some of them deleterious, leading to more restrictive use or abandonment. There are numerous examples, particularly for chronic treatments, such as those employed in diabetes and depression [1,2]. In many cases, the collateral clinical effects are mediated by mechanisms other than those associated with the target for which the drug was initially designed. Collateral clinical effects are often justified by the presence of the initially identified target in another organ, tissue or cell. Both scenarios raise concerns about pleiotropic effects and emphasize the necessity for a comprehensive characterization of the various molecular impacts of the drug, especially those linked to its target. Rather than considering pleiotropy as a more or less beneficial inevitability, we are seeking to highlight the value of using one of its aspects, i.e., targeting a molecular target present in different tissues, to seek convergence toward specifically beneficial effects. Here, we seek to illustrate this point of view by taking stock of knowledge on two pathologies that are often linked: diabetes and depression.

Depression is the most common psychiatric pathology, with a prevalence that is estimated to range from 5% to 20% of the general population [3]. Depressive disorders are characterized by sadness of sufficient severity or persistence that interferes with daily living and often by diminished interest or pleasure in activities (anhedonia). The exact cause is unknown but is probably multifactorial, involving heredity, altered neurotransmitter levels, altered neuroendocrine functions and psychosocial factors. Therapeutic approach usually involves medication and/or psychotherapy. Depressive states are often associated with deficits in serotonin (5-hydroxytryptamine, 5-HT), which is an essential neurotransmitter for communication between neurons and is involved in eating, sexual behavior, the sleep–wake cycle, pain and anxiety or mood disorders [4,5]. Being defined by the World Health Organization as a common mental disorder worldwide, depression is the main mental disability leading to death (WHO, 2021) [6]. Numerous reports suggest that 2/3 of individuals taking antidepressant drugs actually benefit from their medications. However, for the remaining 1/3, antidepressants currently available on the market are ineffective and/or make their depressive symptoms worse [7].

Diet-Induced Obesity (DIO) and type 2 diabetes mellitus (T2DM) also represent major healthcare problems. DIO alone was identified as the cause of 80% of all T2DM cases, and both disorders mainly result from adverse eating habits and inadequate physical activity. Although there is an abundance of research examining the complex association between obesity and major depressive disorder (MDD), the conclusions are still inconsistent [8]. Whereas the larger body of evidence is leaning toward the presence of a link between these two pathological conditions [9], several studies report that they are unrelated [10] or only show an association in subgroups, for example, in women [11]. A review [9] summarizing the epidemiological evidence of the interconnection between obesity and MDD from large meta-analyses suggests overall that obesity and depression are bi-directionally associated, with the presence of one increasing the risk of developing the other. Overall, the rate of mild, moderate and severe depression in patients with diabetes increases with a higher body mass index (BMI). Subjects with obesity and diabetes appear to be at an even higher risk for depression compared to subjects with obesity but not diabetes [12]. Thus, subjects with obesity and diabetes are at greater risk of depression compared to the general population.

We should also mention meta-analyses showing the effects of antidepressant treatments on diabetes in patients with depression. This shows that some antidepressants (escitalopam and agomelatime) have a beneficial effect on glycemic control [13]. Conversely, numerous trials have been conducted to explore the potential of antidiabetic treatments such as metformin, thiazolidinediones and GLP-1 as antidepressants to ameliorate depression [14]. Results suggest that there are shared molecular mechanisms underlying these conditions that we should take advantage of for developing drugs that can effectively treat both conditions, and the authors insist on the need for rationality to guide the tailoring of future treatments. Furthermore, recent reviews emphasize that, given the established link between these two pathologies, common causes or molecular mechanisms must be sought [15].

In this review, a focus on drugs that target excitable cells such as neurons or insulin-secreting beta cells will be made. We take stock of the effects of two reference treatments at central and endocrine levels: fluoxetine for depression and GLP1 receptor agonists for diabetes. To complete the picture, we will show how this pleitropic approach can be implemented in preclinical research by using the example of an experimental antidepressant (spadin and its derivatives) based on a sortilin-derived propeptide (PE) that we recently demonstrated to have beneficial potential in diabetes. However, rather than looking for related mechanisms implicated in the onset of depression and diabetes, we sought to identify whether the molecular target could exert beneficial effects on cell types at the heart of both pathologies, neurons and pancreatic beta cells, and if so, whether this could lead to a synergistic action on comorbidities. To corroborate this point, this review draws parallels between the mechanisms and effects of two gold-standard treatments for diabetes and depression, and the signaling pathways mobilized by the closure of TREK-1 background K+ channels through PE/spadin interaction. As a treatment of choice for depression, we will summarize the effects of fluoxetine on both pathologies. In parallel, the benefit of the drug homologues of the enteroendocrine hormone glucagon like peptide-1 (GLP-1) in diabetes and mood will be described. Finally, we will report the properties of the new class of molecules derived from PE in these comorbidities.

## 2. Depression and Diabetes as Comorbidities

As mentioned above, obesity can increase the risk of depression, and depression is predictive of developing obesity. Psychological stresses frequently lead to modifications of hormone levels and proinflammatory molecules (C-reactive protein and cytokines) that generate a higher risk of type 2 diabetes and depression [16]. Both overall adiposity (total body fat and BMI) and abdominal adiposity (waist circumference and visceral adipose) measures are associated with depressive mood. The strongest association is observed between levels of adiposity and specific “atypical” neurovegetative depressive mood symptoms (e.g., fatigability and hyperphagia), which may be an indication of an alteration in the energy homeostasis. A higher degree of obesity is likely causal for the specific symptom of increased appetite in participants with depression. Indeed, subjects with atypical depression have markedly elevated obesity rates compared to population controls and to other subjects with depression [17]. In contrast, obesity rates are not significantly different in subjects with classic depression and controls without depression. Thus, refining the target phenotype(s) for future work on depression and obesity might improve our understanding, prevention and treatment of this complex clinical problem [18]. There is also several established molecular links between depressive pathology and some adipose-related metabolic signals such as glucocorticoids, leptin, adiponectin, resistin, insulin and inflammatory signals [16]. Elevated glucocorticoids levels, produced by adrenal glands, are implicated in the pathophysiology of both obesity and depression. Indeed, the critical role of corticoids on adipose tissue deposition was demonstrated by studies showing that adrenalectomy prevents obesity [19]. In addition, repeated administration of corticosterone to rodents is reported to display depressive-like behavior [20], and leptin-deficient *ob/ob* mice (obese and hyperglycemic animals) have elevated corticosterone, which is reduced by leptin treatment [21].

Genetic analysis of risk factors for MDD and T2DM almost expectedly shows an association of comorbidity genes involved in natural immunity or cellular aging [22]. Genes relevant to the innate immune system, tau protein formation and cellular aging were identified, and the experimental results indicate that the common, often comorbid, conditions of MDD and T2DM have a common molecular pathway [22]. For example, overexpression of the BDNF gene in the dorsal raphe nucleus (DRN) of obese and diabetic mice subjected to a stress-induced depression protocol will have an associated antidepressant effect by improving serotonin homeostasis [23]. In addition, the improvement in metabolic biological constants due to BDNF overexpression shows, as expected, the importance of the DRN in depression as well as the importance of this brain area in diabetes [23].

Nowadays, various therapeutic approaches are proposed for patients with depression and diabetes. Among these, drugs targeting a specific receptor or channel at the transmembrane level have proven effectiveness, bringing them to the level of major therapeutic approaches such as fluoxetine for treating depression and GLP-1 as an antidiabetic. Moreover, besides their original expected effect, their clinical long-lasting use has revealed unexpected additional beneficial effects.

## 3. Fluoxetine

### 3.1. Fluoxetine as an Antidepressant Reference Treatment

Since the 1960s, the strategies based on antidepressant molecule development have mainly focused on increasing the quantity of 5-HT released in the synaptic cleft, the space between two neurons where nerve communications take place via neurotransmitters. 5-HT can activate the different subtypes of the 5-HT receptor family (1, 2, 3, 4, 5, 6 and 7), leading to their respective signal transduction pathway within the postsynaptic neurons [24]. In presynaptic 5-HT terminals, 5-HT is either taken up by storage vesicles through selective serotonin transporters or degraded by monoamine oxidase (MAO). Some MDD treatments target the serotoninergic system through two pharmacological approaches: selective 5-HT1A receptor antagonists [25] and selective serotonin reuptake inhibitors (SSRIs), including fluoxetine [26]. This latter class of antidepressants is considered serotoninergic because they increase intrasynaptic serotonin concentrations by inhibiting presynaptic 5-HT reuptake, leading to the stimulation of postsynaptic 5-HT receptors. Thus, the serotonin remaining at the synaptic cleft for a longer period of time would repeatedly stimulate the receptors of the postsynaptic cell.

Fluoxetine was discovered in the 1970s. Initially called LY110140, it was described as a selective 5-HT reuptake inhibitor [27,28]. Fluoxetine hydrochloride (better known as Prozac^R^) was the first molecule in the family of antidepressants known as SSRIs [29], the most widely prescribed antidepressants for the treatment of depressive states nowadays. The first clinical study conducted in 1993 showed its efficacy on severe depression with few side effects, allowing its use for long-term treatment [30]. It took several years to demonstrate the physical interaction between the serotonin transporter (SERT) and fluoxetine. Indeed, at the molecular level, SSRIs bind directly to the SERT to maintain the transporter in an outward open conformation, preventing the binding of substrates [31]. SSRIs are selective for the serotonergic 5-HT system but not specific for a particular 5-HT receptor. Indeed, they allow stimulation of 5-HT1 receptors, combining antidepressant and anxiolytic effects, as well as that of 5-HT2, often causing anxiety, insomnia and sexual dysfunction, and 5-HT3 receptors, inducing nausea and headache. Thus, selective serotonin reuptake inhibitors can paradoxically relieve and generate anxiety. In addition, it was shown very recently that antidepressant drugs were binding directly to the TRKB neurotrophin receptor, which facilitates BDNF stimulation [32]. This new piece of knowledge highlights the complex effects of SSRI drugs and strengthens the evidence for their potential pleiotropic action.

### 3.2. Fluoxetine Action on Pancreatic Endocrine Function

In the clinic, it became apparent early on that fluoxetine could be used on very large patient populations. Patients with diabetes are as sensitive to fluoxetine as patients with depression, and this treatment also improves glycemia after only few weeks [33,34,35]. Indeed, fluoxetine tends to improve glycemic regulation and weight loss by inducing higher insulin sensitivity and regulation of skeletal muscle glycogen synthase activity [36]. In fact, this increase in insulin sensitivity seems to be one of the major effects observed among patients treated with Prozac [37]. Fluoxetine significantly reduces food intake in lean or obese rats [38,39], and its indirect effect on weight maintenance is achieved by the balance between food intake and energy expenditure managed by the hypothalamus [40]. Interestingly, an atlas of vagal sensory neurons has recently been published. The authors indicate that serotonin is expressed in specific neuron types [41]. This suggests a possible peripheral effect of fluoxetine on these neurons, which innervate the pancreas. In this vein, in mice, electrostimulation of the pancreatic nerve has been shown to be an effective approach to eradicating recent-onset type 1 diabetes [42].

Interestingly, serotonin is expressed in endocrine cells of the pancreas and is sequestered in the same secretory granules as pancreatic hormones [43,44]. Whereas on one hand 5HT regulates the pancreatic secretion of glucagon and insulin (32), 5-HT secretion is, on another hand, regulated by glucose in β-cell lines (MIN6) in vitro [45] and vesicular transporters of 5-HT (VMAT1/2) are expressed in pancreatic β cells [46]. In addition, the increase in β-cell mass during gestation requires the control of serotonin homeostasis. Thus, altered serotonin signaling also contributes to β-cell mass dysfunction and to diabetes [43]. The clearance transporter SERT is also expressed in β cells [47], suggesting an effect of fluoxetine on the endocrine pancreas. Recently, a direct effect of fluoxetine on pancreatic β cells to potentiate insulin exocytosis has been shown [48], and preliminary experiments on *ob*/*ob* mice show improvement in metabolic physiological parameters [48]. However, several other in vitro studies have shown the opposite results [49,50,51]. For example, insulin secretion is inhibited by fluoxetine [49,51] in rodent and [47] in human islets. Additionally, the increase in serotonin concentration outside the β cells induces the dysfunction of mitochondrial activity, which is by itself coupled to insulin secretion [49]. Thus, despite the extensive research conducted in the field, the effects of 5-HT on the endocrine pancreas remain difficult to grasp because of the expression of various receptors and transporters of 5-HT in the islets of Langerhans [52] (Figure 1). For now, the molecular data allow us to anticipate a long-term effect of treatments using SSRIs, as shown in animal model studies [53]. The complex effects of serotonin on the adaptive mechanisms of the endocrine pancreas suggest that caution is required in the use of drugs targeting this signaling system.

## 4. GLP-1

### 4.1. GLP-1R Agonists as Antidiabetic Drugs

Current front-line treatments for type 2 diabetes are not fully satisfactory because they do not act on weight loss and/or improve β-cell function. Glucagon-like peptide-1 (GLP-1) analogues or GLP-1 receptor agonists are promising treatment options because they improve glycemic control, as well as decrease weight by approximately 2–3 kg/year. In addition, they offer the hope of stabilizing or improving β-cell function by promoting the proliferation or inhibiting the apoptosis of β cells. Thus, GLP-1R agonists represent a good opportunity to treat patients who are inadequately controlled by the classical combination of insulin sensitizer–insulin secretor agents such as metformin–sulfonylurea, and their efficiency can be compared favorably with insulin therapy [54].

Food intake induces a number of physiological adaptations that enable nutrient absorption and metabolism. Among these adaptations, some gastrointestinal hormones, called incretins, facilitate glycemic homeostasis by stimulating insulin secretion. The demonstration of the intestinal–pancreatic axis comes from the observation that a bolus intake of glucose results in a higher insulin secretion than the same amount administered intravenously. The first incretin identified, GIP (glucose-dependent insulinotropic polypeptide), is a 42-amino-acid hormone synthesized by enteroendocrine K cells in the duodenum and jejunum. It weakly inhibits gastric acid secretion and stimulates insulin exocytosis. The second, GLP-1 (glucagon-like peptide-1), is produced from the proglucagon gene and is secreted by enteroendocrine L cells in the distal ileum and colon [55]. The plasma level of GLP-1 is around 5–10 pmol/L in the fasting state and increases rapidly after carbohydrate intake, reaching about 15–50 pmol/L. GLP-1 and GIP are rapidly degraded, with a half-life of two minutes, and are eliminated by two enzymes, dipeptidyl peptidase 4 (DPP4) and neutral endopeptidase (NEP), respectively.

GLP-1 stimulates insulin secretion in a glucose-dependent manner, and its insulinotropic effect is lost when the glucose value is below 4.5 mmol/L. GLP-1 and GIP play an extremely important role in glucose homeostasis since their additive insulinotropic effects are responsible for about 60% of the insulin secreted after a meal in humans. In patients with type 2 diabetes, GLP-1 and GIP secretion are relatively normal, but their ability to stimulate insulin secretion is decreased by about 50% for GIP and 30% for GLP-1 compared to subjects without diabetes. GLP-1 stimulates insulin secretion in β cells but also activates insulin gene transcription, increases insulin biosynthesis, stimulates cell proliferation and survival and decreases cell death. In addition to insulin β cells, GLP-1 acts on other tissues, such as the central and peripheral nervous system, heart, kidney, lung and digestive tract. Finally, GLP-1 inhibits glucagon secretion, slows gastric emptying and increases the feeling of satiety (Figure 2).

The overall effects of GLP-1 on glucose homeostasis as well as on the feeling of satiety and β-cell mass have raised considerable interest in the treatment of type 2 diabetes. Indeed, most antidiabetic treatments act by increasing either insulin secretion (sulfonylureas, glinides) or peripheral insulin sensitivity (metformin, glitazones). But none of them seem to target the two most important parameters in the evolution of type 2 diabetes, body weight and progressive deterioration of the functional β-cell mass, at the same time. Thus, GLP-1, due to its pleiotropic effects, particularly at the level of food intake and β-cell mass, brings hope for the long-term management of this disease [56].

GLP-1′s insulinotropic activity is exerted by its interaction with a specific receptor, namely the GLP-1 receptor, which belongs to the GPCR family. Its binding activates the adenylate cyclase via Gs, resulting in the formation of cAMP [57]. This results in the activation of protein kinase A and cAMP-regulated guanine nucleotide exchange factor II (cAMP-GEFII, also known as Epac2), leading to a plethora of events, including altered ion channel activity, intracellular calcium handling and enhanced exocytosis of insulin-containing granules [58]. The increase in cAMP concentration in the vicinity of the plasma membrane of β cells potentiates glucose-stimulated insulin secretion (GSIS) when the glucose level reaches a certain threshold [59]. It also evokes depolarization of the membrane through its effect on the sulfonylurea-sensitive channels that are responsible for the depolarization process [60]. The inhibition of the adenylate cyclase by Gi is caused by membrane repolarization and a reduction in calcium influx [61]. The potentiating effect of GLP-1 initiate numerous events, including PKA-dependent phosphorylation, which promotes inhibition of K^+^-ATP channels, opening of L-type VOCs (and thus Ca2^+^ influx) and inhibition of voltage-dependent repolarizing K^+^ channels. The complex SUR1-KiR6.2 proteins, the pore-forming subunit of the KATP channel and the alpha1 subunit of calcium channels are PKA substrates [62,63]. Thus, PKA induces an overall increase in Ca2^+^ influx [64,65]. The increase in cAMP is associated with an increased amplitude of L-type calcium currents, in a dose-dependent manner [63,66] (Figure 3).

One of the most important parts of the regulation pathway involves the secretory machinery. Translocation of insulin granules from the reserve pools close to the plasma membrane increases the size of the readily releasable granule pool (RRP) by accelerating their filling rate [67]. In addition to this enhancement of the glucose effects on the so-called proximal steps of exocytosis, PKA-dependent phosphorylation of exocytosis machinery proteins (the SNARE complex) [68,69,70] allows GLP-1 to sensitize the insulin granule docking and fusion complex to calcium influx. The numerous targets of PKA-dependent phosphorylation (CSP, Snapin, SNAP25) [71,72,73,74,75], which are very abundant in neurons, are present in endocrine cells.

### 4.2. GLP-1 as a Neuromodulator

GLP-1 is present in the brain [76] and is involved in the regulation of food intake in the hypothalamus and reward areas [77]. GLP-1 has anorexic effects via direct or indirect activation of nucleus arcuate neurons [78] and activates the POMC and CART neurons through the control of AMPK. One part of the central GLP-1 is produced in the *tractus nucleus solitari* and *medulla vasolateral* [76], and another part derives from the peripheral circulation through the Blood–Brain Barrier (BBB) [79]. GLP-1 analogs and agonists of the GLP-1 receptor (GLP-1R) are able to cross the BBB as well [80]. In addition, in humans, GLP-1R is expressed in the cerebral cortex, hypothalamus and *septum lymbic* [81] with an expression pattern that could suggest a role in mood regulation [82].

In both humans and rodents, GLP-1 injection activates the hypothalamic–pituitary–adrenal (HPA) axis with a resulting increase in both ACTH and corticosterone/cortisol concentrations in the blood. As increased secretion of cortisol is observed in cases of severe depression, especially those associated with anxiety, the fact that GLP-1 may interfere with cortisol level is of interest. Indeed, chronic administration of GLP-1 (12 weeks of treatment with liraglutide) in patients with obesity induces a decrease in urinary cortisol concentration [83]. Although acute and chronic GLP-1 administration lead to different effects on cortisol concentrations, they illustrate the possible modulatory effects of GLP-1 on mood in humans. Interestingly, the behavior-modulating effect of GLP-1 is not limited to its effects on cortisol concentrations but also acts directly on neurons [80,84]. In rodent models, GLP-1R agonists (liraglutide or exenatide) increase anxiety behaviors [85,86], and anxiogenic effects may be associated with an acute increase in corticosterone and ACTH. Conversely, chronic administration of GLP-1 or analogues is anxiolytic [87,88,89]. Thus, GLP-1R is required in animals for adaptive behavior in response to stress. A few human studies have been published on the effects of GLP-1 on depression and anxiety. In 2010, a large cohort study did not reveal any effect of liraglutide on depression or anxiety scores in treated T2D patients [90]. However, in 2011, a following study using another GLP-1 analogue, exenatide, showed decreased anxiety and depression scores in T2D patients [91]. In addition, subjects with bipolar syndromes and depression have found benefit from liraglutide treatment [92]. Indeed, weight loss associated with the improvement in glycemia is conducive to a feeling of well-being [93]. Overall, the main benefit of GLP-1R agonist therapy is an improved quality of life [94].

To conclude this section, while current therapeutic molecules designed to hit a single target in a specific tissue have demonstrated some potent pleiotropic effects, it was a matter of chance whether or not the side effects led to additional beneficial actions. In the field of pharmacology related to diabetes and depression, our recent findings on PE/spadin peptides strongly suggest that the upstream elucidation of the multiple biological actions induced by a single molecular mechanism would enable us to better anticipate pleiotropic effects, especially positive ones.

## 5. PE/Spadin

Several new classes of antidepressants are actually under investigation, i.e., potentially safer peptide antidepressant compounds. The study of antidepressant response in mouse models of depression has allowed the identification of a new set of genes whose association with remission has been examined in a large treatment efficacy trial. The Sequenced Treatment Alternatives to Relieve Depression (Star(*)D study) was conducted in order to decipher the mechanisms of action of antidepressants [95]. One of these identified genes codes was for TREK-1, a neuronal background potassium channel widely expressed in brain areas implicated in major depression [95,96]. Among all the targets identified for the treatment of depression, TREK-1 channels were characterized a few years ago [97,98] as being directly involved in mood disorders. TREK-1 and two other background K^+^ channels, TASK-1 and TASK-2, are activated by volatile anesthetics and may therefore contribute to the central nervous system (CNS) depression produced by these volatile compounds [99]. TREK-1 belongs to the family of two-pore-domain potassium channels (K_2_P) [100]. These channels contribute to the background (or leak currents) that set the resting potential and oppose depolarizing currents. TREK-1 is activated by membrane stretch, volatile anesthetics, acidosis and polyunsaturated fatty acids [101,102,103]. The first phenotyping study conducted on TREK-1-deficient mice demonstrated a major role of the channel in the control of depression [97]. Indeed, several studies have established that these channels are required for mood stability, for example, in a post-stroke model of depression [104]. Recently, it has been shown that an endogenous peptide (PE) of 44 aa, released after the cleavage of prosortilin by furin [105], and its synthetic counterpart spadin have potent antidepressant effects in rodents [106]. Sortilin is a class 1 receptor involved in the sorting of many types of proteins, such as transmembrane proteins (GLUT4, LDLR, p75NTR, NTSR1, TREK-1, etc.) [107]. PE/spadin action comes from its ability to specifically block TREK-1 currents by directly binding the channel with high affinity. The membrane depolarization induced by this blockage leads to the efficient antidepressant action of the peptides in several behavioral models of depression [106]. In particular, spadin increases the activity of 5-HT-secreting neurons and induces neurogenesis with a delay in action of only 4 days [106]. Spadin does not produce any side effects on the functions controlled by the TREK-1 channel (pain, epilepsy, heart function) [108]. It was also demonstrated that spadin directly acts on neurons to trigger the activation of the PI3 kinase pathway, a well-described survival pathway, leading to an increase in spine maturation [109].

Serum PE levels are decreased in patients with major depressive disorder vs. healthy individuals, and these levels are restored to normal after antidepressant treatment [110], as confirmed in cohorts of antidepressant-treated patients. Among patients resistant to pharmacological treatment, those treated with electroconvulsive therapy show a significant increase in serum PE levels one month after therapy [110]. Hypothetically, PE could serve as a marker of depressive state and also as an indicator of remission of the pathology [111]. Interestingly, mice deficient in sortilin show a phenotype similar to TREK-1-deficient mice [112] (Figure 4), suggesting common molecular events driven by sortilin and TREK-1.

While the antidepressant potential of PE/spadin peptides can easily be identified, we wondered whether the same pathway that is expressed in the endocrine pancreas might be of interest in the treatment of diabetes.

Recently, in vitro and in vivo studies have demonstrated that peripheric inflammation, induced by obesity and diabetic status, leads to a decrease in sortilin (which generates PE) expression in adipocytes, skeletal muscles [113] and the liver [114,115]. Additionally, sortilin expression (from *sort1* gene) was inhibited in 3T3L1 adipocytes treated with TNFα [113,116]. Finally, downregulation of sortilin has been also observed in the adipose tissue and skeletal muscles of patients with diabetes [117]. Sortilin and TREK-1 channels are both expressed in pancreatic beta cells [118,119,120], and TREK-1 channel closure induced by spadin potentiates glucose-dependent insulin secretion in response to a glycemic challenge [120]. Spadin acts on the plasma membrane potential in a similar manner to exendin-4, a potent GLP-1R agonist, and amplifies glucose-induced (20 mM) plasma membrane depolarization [121]. In vivo, spadin is able to improve the recovery of mice during an intra-peritoneal glucose tolerance test (IPGTT) [120] by potentiating insulin secretion. The circulating insulin concentration in spadin-injected mice is significantly higher. In contrast, spadin has no effect on glucose storage in adipocytes [116]. Given the properties of PE on neurons, it was postulated that PE and its synthetic derivatives could protect pancreatic β cells from dysfunction and death. Results showed that modulation of membrane potential has a protective effect on both endocrine cells and neurons [106,109,122]. Indeed, PE and its derivative spadin have anti-apoptotic, proliferative and maintenance of function effects. This protective pattern is possible as PE/spadin application promotes a rise in intracellular calcium concentration [123] that activates the calcium/calmodulin Kinase (CAMK-2 and 4) [122] and thus activates Akt and ERK survival and proliferative pathways. In addition, the level of P-CREB in the nucleus increases in the presence of PE/spadin, especially under cellular stress. Interestingly, P-CREB immunoreactivity was detected in the hippocampal neuron nucleus in spadin-treated mice [106], suggesting that similar molecular processes are activated in both neurons and beta cells (Figure 4).

Unexpectedly, it was shown that the antidepressant fluoxetine (Prozac^R^) is as able as PE and derivatives to act directly on TREK-1, producing a subsequent concentration-dependent inhibition of TREK-1 current [124]. Structural tridimensional studies performed on crystallized TREK-2 channels demonstrated state-dependent inhibition of TREK channels by Prozac^R^ [125]. Fluoxetine’s inhibition of TREK channels could therefore have similar beneficial effects on both neurons and beta cells.

## 6. Concluding Remarks

This review supports the concept of searching for new therapeutic targets by identifying proteins involved in the same pathway in different cellular systems, which could offer multiple positive effects. In this approach, the “side effects” cooperate. Thus, the long-term effects of SSRIs are difficult to assess, but it is known that serotonin has a complex role in the periphery, for example, in the endocrine pancreas. It is therefore not possible to eliminate all effects on long-term pathologies such as diabetes. However, the use of a drug that targets a signaling pathway through a specific receptor is more effective in terms of positive effects. GLP-1 is a good example, because even if most of its positive effects are peripheral, there are enough neuronal pathways sensitive to its agonists to generate beneficial effects at the central level on eating behavior and mood control. This emphasizes once again that the periphery is able to modulate the central activity of neurons. Finally, our recent work has shown that modulation of the membrane potential, through the closure of background potassium channels, systematically generates facilitating effects on the exocytosis of hormones and neurotransmitters as well as on the activation of survival pathways (Figure 4). These two parameters are key to achieving an optimized cure for chronic diseases such as depression and diabetes.

Finally, it seems that our current knowledge should enable us to better anticipate some pleiotropic effects and develop drugs that can effectively serve in multiple conditions, specifically comorbidities like depression and diabetes. In fact, we have at our disposal the distribution of gene expression (genomics and transcriptomics) and protein expression (proteomics) in numerous tissues. The use of this knowledge, together with the characterization of molecular targets, will be decisive for better drug positioning and could lead to more efficient and integrated therapeutic solutions for individuals facing multiple health challenges.

## Figures and Tables

**Figure 1 cells-12-02768-f001:**
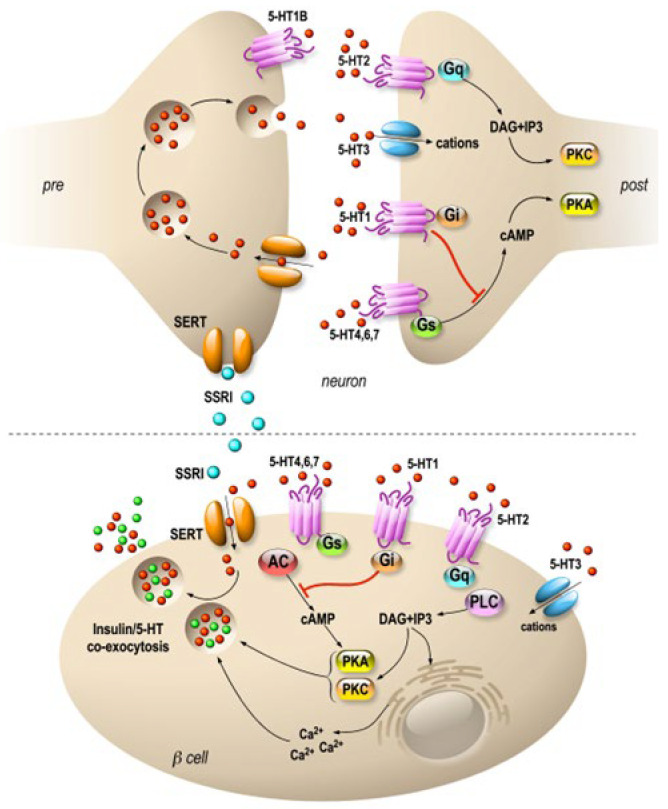
Specific molecular targets of fluoxetine on neurons and pancreatic β cells. Distribution of 5-HT transporters and receptors in neurons (upper panel) and in pancreatic β cells (lower panel). In neurons, 5-HT receptors are expressed mostly at the postsynaptic level, and they modulate signal transmission. β cells express 5-HT receptors similar to those of neurons. SSRIs are symbolized by blue, 5-HT by red and insulin by green dots. 5-HT1, 2, 4, 6 and 7 are 7-TM domain receptors (GPCRs). 5-HT3 is a cationic channel symbolized by blue. SERTs are in orange.

**Figure 2 cells-12-02768-f002:**
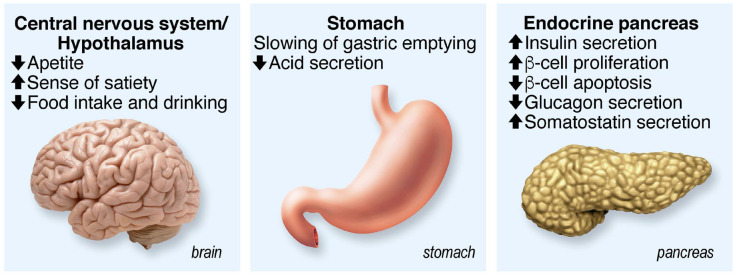
Summary of GLP-1 functions. Acting on several physiological functions, GLP-1 is a multitask hormone that regulates general body metabolisms.

**Figure 3 cells-12-02768-f003:**
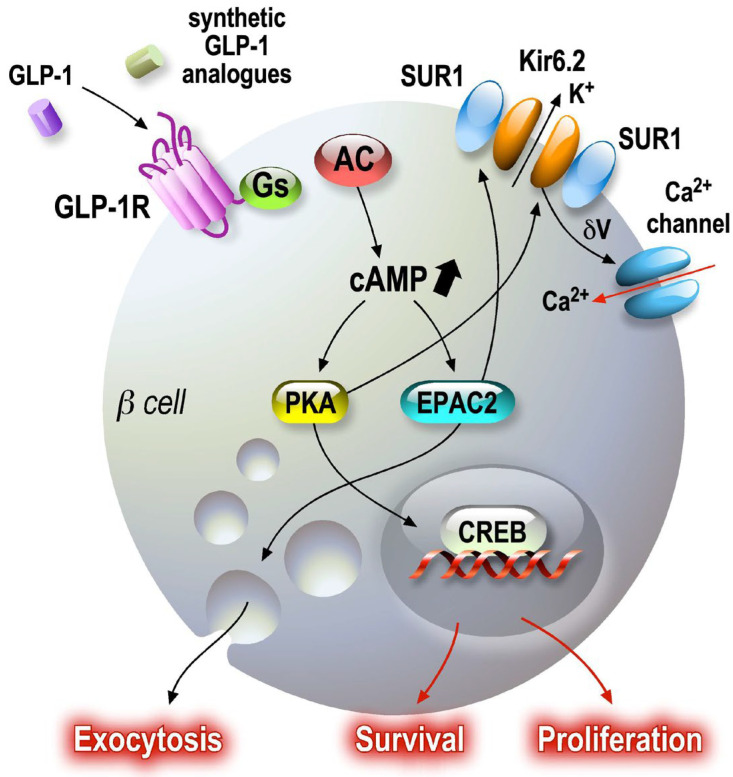
GLP-1 signaling pathways in pancreatic beta cells. Secreted GLP-1 from enteroendocrine L cells in the small intestine epithelium acts on several target cells, such as pancreatic beta cells. GLP-1 receptor agonists are associated with cAMP-dependent pathways, which amplify regulated exocytosis and increase cell survival and proliferation. Amplification and survival pathways controlled by cAMP as a second messenger activate PKA and EPAC sensors. Once activated, these sensors maintain endocrine function by modulating CREB transcriptional activity.

**Figure 4 cells-12-02768-f004:**
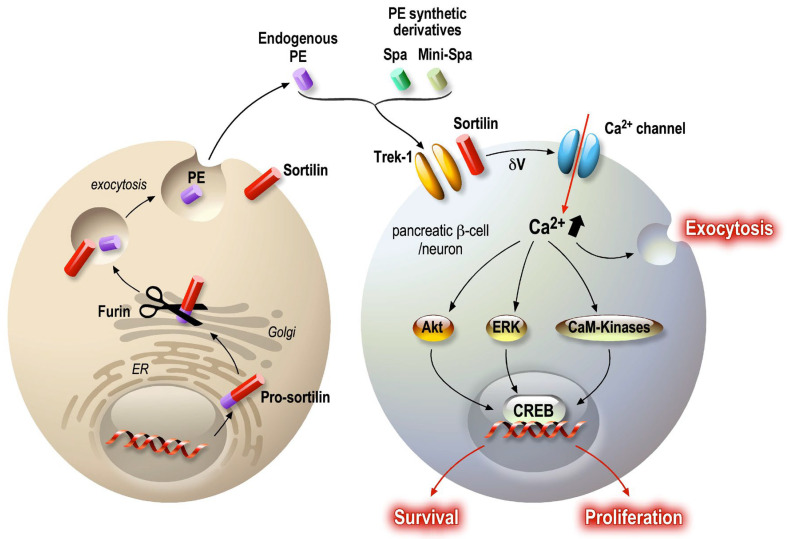
PE signaling pathway in neurons and pancreatic beta cells. PE is present in the general circulation, being secreted by several cells, such as adipocytes, skeletal muscle and the intestinal epithelium. PE is a specific K_2_P TREK-1 channel blocker, thus inducing partial plasma membrane depolarization. The subsequently induced calcium entry activates several signaling pathways: Akt, ERK and CaM-Kinases. The final resulting activation upregulates CREB transcriptional activity, maintaining cellular functions. As described, secreted peptides can modulate central and peripheral targets and provide co-benefits for different but related pathologies. In both cases, the target is a common molecular system that generates similar protective and functional improvements for the challenged organs.

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
