# Peer review of "Unlocking Therapeutic Synergy: Tailoring Drugs for Comorbidities such as Depression and Diabetes through Identical Molecular Targets in Different Cell Types"

_cells, 2023, doi:10.3390/cells12232768_

Round 1
Reviewer 1 Report
Comments and Suggestions for Authors
The content of this manuscript could be interesting. However, several cahnges should be done to improve clarity.
A) The aim of the manuscript should be clear.
B) The pleiotropy of drugs must be clearly presented in introduction. There are several "clasic" examples reported and reviwed.
C) Authors should revise current literature about its proposed examples. Then, please choice and edit your manuscript to present the best proposal for applying pleiotropism of drugs and the shared mechanism(s) in some diseases. For example: in the first proposal, the following paper https://www.ncbi.nlm.nih.gov/pmc/articles/PMC9421708/ shows related data which could limit the proposal in your manuscript (fluoxetine may reduce weight by −2.7 kg (95% CI −4 to −1.4; p < 0.001) and body mass index by −1.1 kg/m2 (95% CI −3.7 to 1.4), compared with placebo; however, it would cause approximately twice as many adverse events, such as dizziness, drowsiness, fatigue, insomnia, or nausea). Please check the potential application in each case and the current knowledge.
D) The most of references support the 'multitarget potential of some drugs'. However, poor relationship is regarding the pleiotropric potential of those drugs. This should be clearly noted.
E) In the conclusions section you sentenced "...pleiotropic approach is more effective than empiricism". However is unclear the comparisson or evidence supported this sentence. Please check and edit it.
Comments on the Quality of English LanguageMinor grammar mistakes could be avoided. Clarity of manuscript could improve after edition.
Reviewer 2 Report
Comments and Suggestions for Authors
The manuscript approaches the literature based on the treatment of depression and diabetes as comorbidities. The idea is relevant and aimed to possible treat the common biological/symptomatological abnormalities observed in these two diseases. However, in my opinion, the parallel between these two diseases was not well constructed, and the development of the sessions based on the available drugs to treat all of these conditions is not clearly presented. It is relevant to mention that this thematic is worthy to be discussed but, at the same time, actually highly published in the literature. For this reason, the manuscript did not have a clear identity, since the drugs choose as possibility to treat all of these conditions are only two: fluoxetine and GLP-1 analogs, and it is not showing a completely new panel of the literature. On the other hand, authors are summarizing an innovative strategy to treat depression and diabetes as commorbidities: PE/spadin. This new strategy is relevant but is not fitting well with commercially available drugs... at least the pathway to put it all together in the MS was not well consctruted. It would be interesting to rise up other innovative strategies to conduct a parallel with PE/spadin, for example. Other strategy to improve the quality of the MS could be rise up all the benefit and limitations of the two drugs choose – fluoxetine and GLP-1 analogs (that could be listed in tables or in schemes), to give the opportunity to the readers to quickly compare the available drugs with innovative strategies, such as PE/spadin and possible phytodrugs.
More specific comments are listed below:
Page 1, 1st paragraph: The neurobiological bases of depression are too poorly explored. Authors mostly resumed depression to the deficiency of monoamines.
Page 2, 2nd session: “multitargeting molecules” – In my opinion, this session is completely non sense, and it is not helping to the essence of this MS. It seems that they have written this paragraph and just put it there.
When the authors talk about the fluoxetine, they completely forgot to mention the clinical trials that investigated diabetes in depressive patients in treatment with this drug. They absolutely should explore this information in the MS.
Considering metformin and PPAR gamma agonists as a highly prescript drug for diabetes worldwide, and the availability of clinical information of these drugs in patients with depression, why authors did not mention these drugs?
The references used in the MS are appropriate; however I listed below some related and relevant references on the field:
Moulton CD, Pickup JC, Ismail K. The link between depression and diabetes: the search for shared mechanisms. Lancet Diabetes Endocrinol. 2015 Jun;3(6):461-471. doi: 10.1016/S2213-8587(15)00134-5. Epub 2015 May 17. PMID: 25995124.
Srisurapanont M, Suttajit S, Kosachunhanun N, Likhitsathian S, Suradom C, Maneeton B. Antidepressants for depressed patients with type 2 diabetes mellitus: A systematic review and network meta-analysis of short-term randomized controlled trials. Neurosci Biobehav Rev. 2022 Aug;139:104731. doi: 10.1016/j.neubiorev.2022.104731.
Grigolon RB, Brietzke E, Mansur RB, Idzikowski MA, Gerchman F, De Felice FG, McIntyre RS. Association between diabetes and mood disorders and the potential use of anti-hyperglycemic agents as antidepressants. Prog Neuropsychopharmacol Biol Psychiatry. 2019 Dec 20;95:109720. doi: 10.1016/j.pnpbp.2019.109720. Epub 2019 Jul 25. PMID: 31352032.
Woo YS, Lim HK, Wang SM, Bahk WM. Clinical Evidence of Antidepressant Effects of Insulin and Anti-Hyperglycemic Agents and Implications for the Pathophysiology of Depression-A Literature Review. Int J Mol Sci. 2020 Sep 22;21(18):6969. doi: 10.3390/ijms21186969. PMID: 32971941; PMCID: PMC7554794.
Reviewer 3 Report
Comments and Suggestions for Authors
Dear Thierry Coppola with co-authors,
I found your review "The value of pleiotropy in tailoring approaches to treat comorbidities such as depression and diabetes" as very timely and comprehensive in the field of neuroscience and endocrinology. It is very important to point towards the peripheral nervous system in the regulation of brain' functions since its role was not appreciated and often overlooked.
The review is well-written, with beautiful illustrations of molecular mechanisms linking 5HT system with pancreatic b-cells, functions of GLP-1 with its signaling and sortilin /spadin as a new molecular drug-target for both depression and diabetes.
I think this review would be improved by adding more information about a role of the 5HT system in the PNS, keeping in mind e.g. stimulation of pancreas via vagal nerve. It will logically link pathological mechanisms on the physiological and molecular-biochemical levels with depression and diabetes. Interestingly, recent study precisely characterized vagus nerve on the molecular level ( doi: 10.1016/j.celrep.2019.04.096) . Moreover, group of Dr. Kupari generated the database, where you can find expression of 5HT genes in vagus nerve : ernforsgroup.shinyapps.io/vagalsensoryneurons/
Round 2
Reviewer 1 Report
Comments and Suggestions for Authors
The changes improve the manuscript. Title is long. Pleiotropism is explained in a improved form. Now, relationship among used examples is notorious. Some proofreading could improve linkage among sentences and clarity of the central topic and used examples. Conclusions should declare clearly scope and limitations of the content of this review.
Comments on the Quality of English LanguageMinor punctuation and grammar mistakes detected.
Reviewer 2 Report
Comments and Suggestions for Authors
The MS improved significantly after revision.